

# A note on the effects of epidemic forecasts on epidemic dynamics

Nicholas R. Record[1,*] and Andrew Pershing[2,*]

[1] Bigelow Laboratory for Ocean Sciences, East Boothbay, ME, USA
[2] Gulf of Maine Research Institute, Portland, ME, USA
* These authors contributed equally to this work.

## ABSTRACT

The purpose of a forecast, in making an estimate about the future, is to give people information to act on. In the case of a coupled human system, a change in human behavior caused by the forecast can alter the course of events that were the subject of the forecast. In this context, the forecast is an integral part of the coupled human system, with two-way feedback between forecast output and human behavior. However, forecasting programs generally do not examine how the forecast might affect the system in question. This study examines how such a coupled system works using a model of viral infection—the susceptible-infected-removed (SIR) model— when the model is used in a forecasting context. Human behavior is modified by making the contact rate responsive to other dynamics, including forecasts, of the SIR system. This modification creates two-way feedback between the forecast and the infection dynamics. Results show that a faster rate of response by a population to system dynamics or forecasts leads to a significant decline in peak infections. Responding to a forecast leads to a lower infection peak than responding to current infection levels. Inaccurate forecasts can lead to either higher or lower peak infections depending on whether the forecast under-or over-estimates the peak. The direction of inaccuracy in a forecast determines whether the outcome is better or worse for the population. While work is still needed to constrain model functional forms, forecast feedback can be an important component of epidemic dynamics that should be considered in response planning.

Corresponding author
Nicholas R. Record,
nrecord@bigelow.org

## INTRODUCTION

The field of forecasting is growing rapidly to include many systems (*Payne et al., 2017*). This growth has magnified many ethical considerations around forecast development, such as conflicts of interest, representation of uncertainty, equity for end users, and unintended consequences, to name a few (*Hobday et al., 2019*). One challenge that is becoming more common is that forecasts increasingly deal with systems that have human components. In conventional forecasts, such as weather forecasts, human responses do not measurably affect the system in question (Fig. 1A). There are more and more cases, however, where human actions are coupled to the system dynamics. Examples include forecasts used in living resource management (*Tommasi et al., 2017*), endangered species

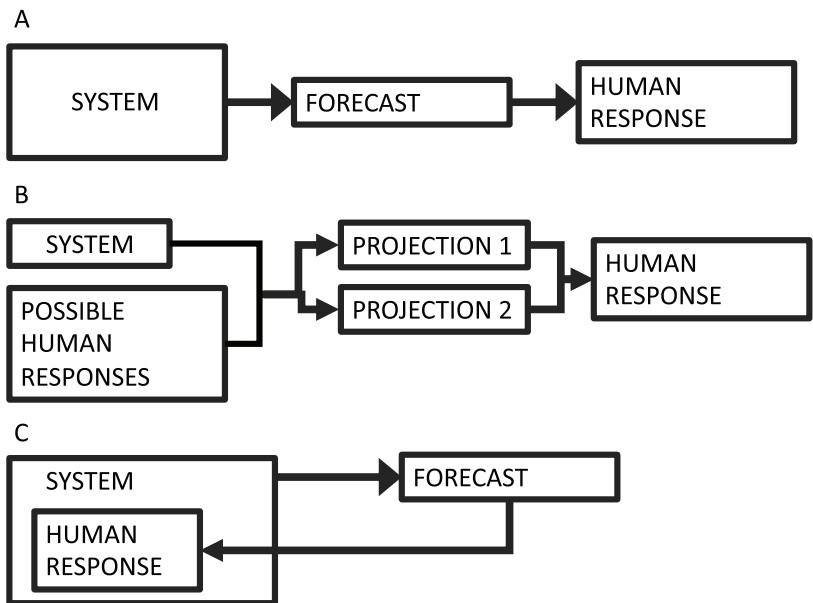

**Figure 1 Forecasting schematic diagram.** (A) The conventional forecasting scheme, where a system informs a forecast, which informs some human response. (B) The conventional scheme for projection, where a range of possible actions produce multiple scenarios (i.e., projections), which can inform action. (C) The forecasting scheme where the human response is part of the system dynamics—that is, forecast feedback or reflexive prediction.

management (*Pendleton et al., 2012*), and epidemic forecasts (*Adam, 2020*). Historically, cases like these have often been framed as projections, which present scenarios that are contingent upon different human choices (Fig. 1B).

In contrast to projections, forecasts provide specific predictions about future events. If humans are changing their behavior in response to information in the forecast, and if that change in behavior alters the outcome of the forecast target, then the forecast is no longer outside of the system, but is instead a dynamically connected component of the system (Fig. 1C). This phenomenon, sometimes refered to as "forecast feedback" or "reflexive prediction" has been historically considered to some degree in economic and political forecasting (*Galatin, 1976*), but is rarely considered in biological systems. According to the "Law of Forecast Feedback", this type of dynamic can hypothetically produce a recursion paradox, where predictability breaks down entirely (*Smith, 1964*). The complexity of dealing with forecast feedback or reflexive prediction is a subject of scientific debate, and it's not clear whether a reflexive forecast could ever be verified empirically (*Kopec, 2011*). As forecasts are increasingly used within the systems they are forecasting, we need to examine the effects that forecasts can have on the dynamics of the system in question.

Epidemic models represent a case where forecasts and human behavior can be very tightly coupled. A forecast predicting a high infection peak can motivate measures like quarantine or social distancing. These measures change the forecasted dynamics, ideally lowering the predicted peak. A less severe outlook can then motivate an easing of measures, which in turn can raise the peak again. This two-way feedback illustrates the difficulty in forecasting systems coupled to human dynamics.

Epidemic models are conventionally used for projections, rather than for forecasts. They can estimate the consequences of certain actions, potential dangers, or scenarios. However, in the modern information-rich environment, there is a demand for real-time forecasts for both management and general audiences, and such forecasts of epidemics are increasingly common. Many of the modeling studies that have responded to the COVID-19 outbreak are forecasts (*Murray, 2020*; *Roosa et al., 2020*; *Fanelli & Piazza, 2020*). The United States Center for Disease Control has even issued an open call for COVID-19 forecasts through a forecasting challenge (https://github.com/cdcepi/COVID-19-ILI-forecasting), and many research teams and news sites serve forecasts to the public in real time. This raises the question: what is the effect of a forecast on the epidemic dynamics? In most biological forecasts, this type of feedback is not explicitly taken into account.

We examined this question using the standard susceptible-infected-removed (SIR) epidemic model. The SIR framework underlies many projections and forecasts that form the basis for response strategies for COVID-19 and other epidemics (*Adam, 2020*). We built forecast feedback into the SIR model by allowing the contact rate to change dynamically in response to different information provided by the SIR equations. In turn, the SIR model dynamics depend on the contact rate, so that there is two-way feedback between the forecast and the other system dynamics. The purpose here is not to fully analyze or characterize the global properties of a particular dynamical model, as there is a whole class of well characterized versions of the SIR model that consider a wide range of additional processes and details (*Beretta & Takeuchi, 1995*; *Anderson & Robert, 1979*; *Takeuchi, Ma & Beretta, 2000*; *Satsuma et al., 2004*; *Kyrychko & Blyuss, 2005*; *Batista, 2020*) that one could modify to explore these feedbacks. Rather, we use simulations of a few simple SIR modifications to illustrate important implications of these feedbacks. Examining a system in this way provides a critical analysis of the role of a forecast in a rapidly changing coupled human system.

## METHODS

Our objective is to test the potential effects of an epidemic forecast on the epidemic itself. The approach is to explore this question in its simplest form. To that end, we use the standard SIR model:

$$\frac{dS}{dt} = -\alpha SI$$

$$\frac{dI}{dt} = \alpha SI - \beta I$$

$$\frac{dR}{dt} = \beta I$$

Here, $S$ is the number of individuals in a population that are susceptible to infection, $I$ is the number of infected individuals, and $R$ is the number of removed individuals (interpreted as recovered/immune, dead, or otherwise removed); $\alpha$ is the contact rate, and $\beta$ is the removal rate. The SIR model has been one of the primary tools for making simulations on which control and intervention measures are based, such as social

distancing, and shelter in place measures, though other statistical and machine learning tools are also used. There are many ways to modify this system of equations to better approximate a particular epidemic. Our interest is in the general effect that model output (either current or forecasted) can have on the dynamics themselves, as mediated through changes in human behavior. The contact rate $\alpha$ is what would likely change with the suggested response measures. A high infection or a severe enough forecast are what motivates measures whose aim is to reduce the contact rate. We therefore test the effect of allowing $\alpha$ to vary as a function of the SIR model output, in a way that is dynamically linked to the other model equations.

Our examination of this system asks two general questions:

1. What if adoption of response measures depends on the current level of infection? This simulates the scenario where the human response is different depending on whether the level of infection is low or high. In this case, $\alpha$ depends on $I$ at any given time.
2. What if adoption of response measures depends on the forecasted peak infection? Peak infection is a primary concern with major epidemics because of health care capacity. In this case, $\alpha$ depends on the predicted peak infection $I_*$, with the prediction made at any given time using a standard SIR solution.

Beginning with case (1), $\alpha$ is a dynamic function, dependent on $I$. We add a fourth equation to our dynamical system, so that the full set of equations includes a dynamical equation for $\alpha$:

$$\frac{dS}{dt} = -\alpha SI$$

$$\frac{dI}{dt} = \alpha SI - \beta I$$

$$\frac{dR}{dt} = \beta I$$

$$\frac{d\alpha}{dt} = r\alpha(1 - \alpha/K)$$

The fourth equation is formulated so that $\alpha$ approaches an asymptotic contact rate $K$. This is a common functional form in ecological theory (*Record, Pershing & Maps, 2014*). There are likely multiple possible functional forms for $K$. There is some evidence that the contact rate is responsive to infection dynamics in an approximately monotonic way. For example, when government response indices (*Hale et al., 2020*) are taken as a proxy for contact rate, and the infection level is scaled to the maximum for each country, a negative relationship appears (Fig. 2A). To make the asymptotic contact rate responsive to the infection dynamics, $K$ is a function of $I$:

$$K(I) = \frac{\alpha_0 - \alpha_*}{N^2}I^2 + \alpha_*$$

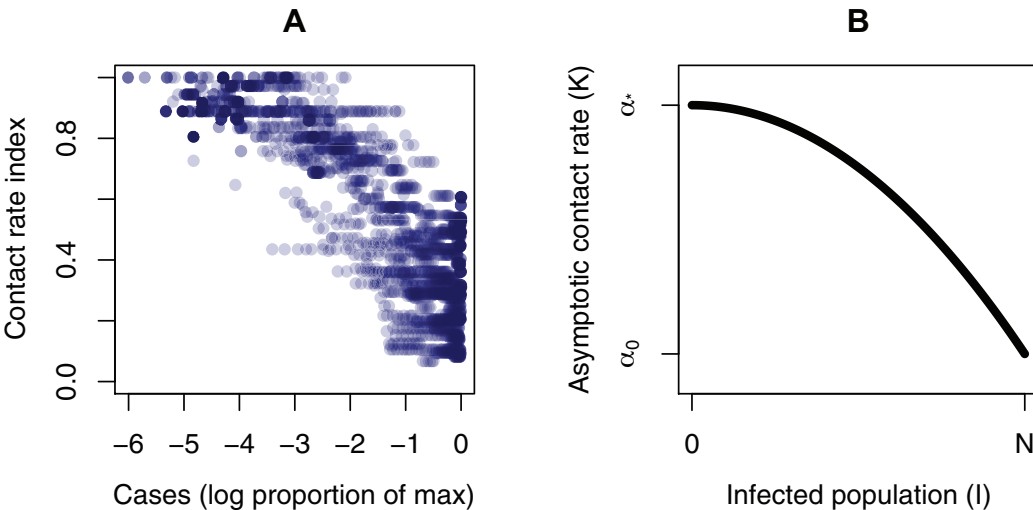

**Figure 2** **Plots informing the model structure.** (A) Evidence of a relationship between the magnitude of infection and a contact rate index. The contact rate is the inverse of the government response index from *Hale et al. (2020)*, so that a higher index corresponds to a higher contact rate. Infection magnitude is scaled to the maximum for each country. (B) The dependance of the asymptotic contact rate ($K$) on the population size ($I$).           

The two parameters $\alpha_*$ and $\alpha_0$ are respectively the unhindered contact rate, and the reduced contact rate after response measures are in full effect. This gives a declining curve in asymptotic contact rate $K$ as a function of the infection level $I$ (Fig. 2B). Finally, we have the parameter $r$, which determines how quickly people adopt response measures—that is, how quickly they move from one $\alpha$ to another. It is likely that from an empirical point of view, $r$ could have a wide range of values, depending on the community and other contextual information. There are other possibilities for the shape of this curve and for the $\alpha$ equation, with different functional forms and producing a wide range of possible dynamics. The quadratic gives a simple curve where the response gets more intense as infections get higher, which allows us to simulate some simple but realistic properties. We use the model in normalized form to show general results, rather than those specific to a particular outbreak (Parameter values: $\alpha_* = 1$, $\alpha_0 = 0.2$, $\beta = 0.1$; Initial conditions: $S = 0.999$, $I = 0.001$, $R = 0$, $\alpha = 1$). With the equations normalized, the total population size is always $N = 1$. As a diagnostic, we use $I_{max}$, the magnitude of the peak infection as the dynamics play out, because of its relevance to health care capacity, as in the COVID-19 pandemic.

Question (2) requires having a forecast. For this, we use an analytical solution for the infection peak (*Weiss, 2013*) of the SIR model,

$$I_* = \left[ 1 - \frac{\beta}{\alpha N} - \frac{\beta}{\alpha N} \log \frac{\alpha N}{\beta} \right] N$$

where $I_*(t)$ is the predicted maximum $I$ value—that is, a forecast of the magnitude of the infection peak, where the forecast is made at any time, $t$. Here, we suppose that people are reacting to the forecasted peak, rather than the current level of infection, and that the

forecast is based on current information at any given time. The forecast solves the standard SIR model, using the current parameter values, as a forecaster would do in real time. This is modeled by replacing $I$ with $I_*$ in the equation for $K$. Again, we use the realized $I_{max}$ as a diagnostic.

The objective here is to illustrate potential dynamics of the feedback between forecast output and human behavior, rather than to ananlyze the global properties of this particular formulation. To this end, we step through simulations where we adjust the human response rate parameter $r$ and compare the diagnostic $I_{max}$. Each simulation represents a different value for $r$. Additionally, the forecasted $I_*$ value at any time represents a hypothetical at time $t$, because the population is constantly responding, and so the forecast is changing. Moreover, as with any forecast, there is the potential that the prediction is an over-or under-estimation of what will happen, due to stochasticity and other forms of uncertainty. Forecast error can potentially affect forecast feedback dynamics. We test this effect by scaling the forecasted $I_*$ value proportionally, from 0.4 to 1.6, and looking at the effect on the infection peak $I_{max}$ across a range of $r$ values. Here, each simulation represents a different accuracy offset.

## RESULTS

We evaluate question (1) by looking at how $I_{max}$ depends on $r$ (Fig. 3A). Responding to the current level of infection leads to a drop in the infection peak (approximately 8% using this set of parameters), as compared to no response, if the response rate is sufficiently fast. If the response rate is very slow, there is effectively no reduction in the infection peak. There is a sharp transition between these two cases. In this example, a response rate slower than about $r = 0.1$—corresponding to a response time of about 10 time units—makes no notable change in the infection peak. For different parameter values, the threshold levels can be different, but the basic shape of these relationships is consistent across different parameterizations.

When the contact rate is responsive to the forecasted infection peak rather than to the instantaneous infection level (question 2), the shape of the curve is similar (Fig. 3A). The reduction in the peak infection is larger, (approximately ~12% drop using this set of parameters, or roughly a 50% improvement over the former case). There is also a notable difference in the needed response time. In this case, a response time of 10 time units ($r = 0.1$) is sufficient for most of the drop to occur. Across parameter space, the magnitudes of these differences change, but the pattern is consistent.

We tested the effects of an over-or under-estimate forecast (Fig. 3B), with the forecasted value $I_*$ scaled proportionally in the range 0.4–1.6. If the forecast is an overestimate, and if the response rate is fast enough, an inaccurate forecast can lower the peak even more than an accurate forecast. An overestimated forecast also gives a larger window of time for response—that is, the response rate does not need to be as fast to see a lowering of the peak. If the forecast is an underestimate, however, it leads to a higher peak and narrows the response time window. Finally, if the response rate is very slow, inaccuracies in the forecast in either direction make very little difference. In all of these cases, responding to a

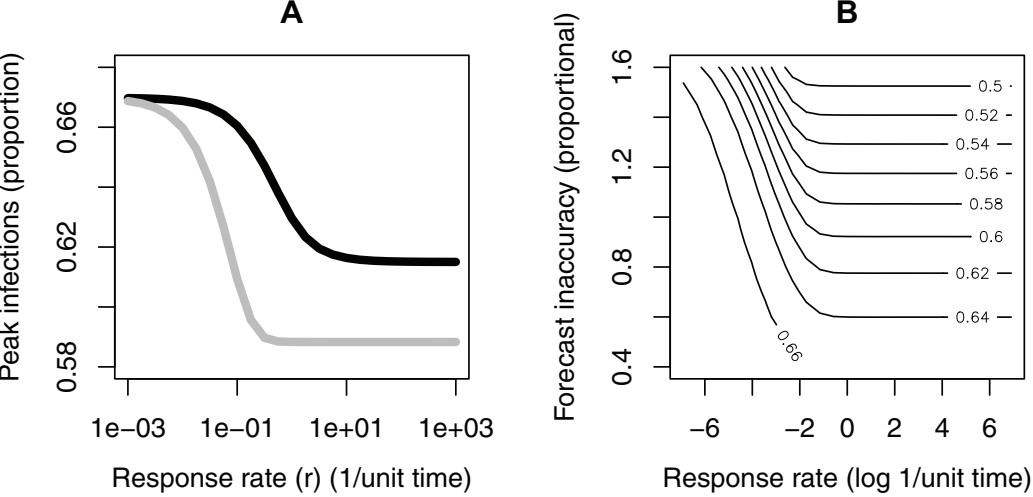

**Figure 3** **Results of model analysis.** (A) The dependance of the infection peak magnitude on the rate ($r$) at which the population reduces the contact rate (i.e., adopts response measures). The population responding to current infection levels (black); the population responding to forecasted infection peaks (grey). (B) The dependance of the infection peak levels (contours) on the rate ($r$) at which the population reduces the contact rate (e.g., adopts social distancing behaviors) and the accuracy of the forecast. In this simulation, the population is responding to forecasted peak infection levels.

forecast is better than not responding at all, due to how we have formulated $K(I)$ as a decreasing function of $I$.

## DISCUSSION

Many new forecasting products are coming online. With the COVID-19 pandemic in particular, studies and online tools with real-time or near-real-time preditions and projections are produced and available rapidly (*Biswas & Sen, 2020*; *Batista, 2020*; *Chen, Lu & Chang, 2020*; *Simha, Prasad & Narayana, 2020*; *Teles, 2020*). People are generally not accustomed to interpreting forecasts as being dynamically linked with their own behavioral responses to the forecast, but such cases are more and more common, and include epidemics, climate change and endangered species management, to name a few.

One growing challenge is the blurring lines between forecasts and projections. For COVID-19, early warnings of a potential global pandemic did not motivate early responses in many places, allowing the virus to move from a local to a global problem. In places like the US and UK, significant action at the national level did not take place until after Imperial College released projections that millions would die in the absence of social distancing (*Adam, 2020*; *Dyer, 2020*), and after signs of the epidemic were widespread. Once actions were taken, the true trajectory began to deviate from the worst case scenarios, and the projections no longer matched reality. This is typical of projections, which aren't intended to be specific predictions, but rather play out different scenarios that depend on initial actions. Climate projections are similar, calculating future

conditions based on Representative Concentration Pathways (RCPs) (*Taylor, Stouffer & Meehl, 2012*). The RCPs each depict a scenario of future carbon such as aggressive emission reductions (RCP2.6) or business as usual (RCP8.5). There is no weighting placed on which pathway society is likely to select. The goal of the projections is to evaluate the consequences of actions and to contrast action with inaction.

Forecasts, on the other hand, do represent specific predictions. We are more accustomed to weather forecasts, where human responses do not change the outcome. Many epidemic models are now offering forecasts (*Murray, 2020*; *Roosa et al., 2020*; *Fanelli & Piazza, 2020*), and in systems like these, humans are a dynamic component, responding to ongoing changes. The human actions depend on the state of the system and the information in forecasts. In reality, not all scenarios are equally available—the likelihood that humans will act depends on the state of the system and the information in forecasts. Examining this dynamic within an SIR model framework illustrates some significant points.

First, as might be expected, using information from a forecast can reduce the magnitude of the infection peak. Additionally, using the forecast also grows the window of time in which a response has an effect. This could give more room to allow for missteps during times when uncertainty is high. Second, inaccuracies in the forecast have an asymmetrical effect on the outcome. An overestimate in the forecast can improve the outcome, in this case lowering the infection peak, but an underestimate in the forecast can actually make things worse. By underestimating the potential danger, response is diminished. Similarly, if the response rate is too slow, a forecast makes very little difference, but an over-estimate forecast can buy more response time. These asymmetries raise the difficult ethical question of whether it is better to give an accurate forecast if a more dire forecast could actually motivate a stronger response. This question becomes even more complex if the forecast is one member in an ensemble that is used for decision making, or is one member in a sequence of forecasts where trust in the forecast is cumulative. If the goal of a forecast is to alter human behavior, then a self-defeating forecast might actually be desirable (*Sabetta, 2019*). In this broader view, accuracy should be considered only one part of a forecast's performance.

## CONCLUSION

We have to be cautious of using forecasts that do not take into account the effects of forecast feedback on the system dynamics. This is especially important in systems where human health depends on the forecast, as is the case with SIR and other epidemic models. The examples shown here represent just one possible formulation for this type of dynamic. Future work is required to better constrain the possible functional forms, both empirically and theoretically. There is a whole class of models with forecast feedback modifications to examine—nearly any dynamical model where human decision making potentially plays an integrated role—producing a wide range of model behaviors. Forecast feedback creates certain challenges and limitations for predictions, possibly paradoxical, but at a minimum forecasters should try to understand the effects this feedback can have on the systems we are trying to predict.

## ACKNOWLEDGEMENTS

We would like to acknowledge the institutions around the world that are working hard to make epidemic data and forecasts freely available. We thank the editor and two reviewers for helpful comments.

### Funding
Support for this research came from institutional funds from the Bigelow Laboratory for Ocean Sciences and Gulf of Maine Research Institute. The funders had no role in study design, data collection and analysis, decision to publish, or preparation of the manuscript.

### Grant Disclosures
The following grant information was disclosed by the authors:
Bigelow Laboratory for Ocean Sciences and Gulf of Maine Research Institute.

### Competing Interests
The authors declare that they have no competing interests.

### Author Contributions
- Nicholas R. Record conceived and designed the experiments, performed the experiments, analyzed the data, prepared figures and/or tables, authored or reviewed drafts of the paper, and approved the final draft.
- Andrew Pershing conceived and designed the experiments, authored or reviewed drafts of the paper, and approved the final draft.

### Data Availability
All files used in this study, and needed to reproduce the results, are available in GitHub: https://github.com/SeascapeScience/SIRforecastfeedback.

### Supplemental Information
Supplemental information for this article can be found online at http://dx.doi.org/10.7717/peerj.9649#supplemental-information.

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
