# Peer review of "A note on the effects of epidemic forecasts on epidemic dynamics"

_PeerJ, doi:10.7717/peerj.9649_

## Round 0.1 · original submission · Major Revisions

Overall, the note has some novelty to it, its not the first instance where modification of society’s behavior changes model predictions, however incorporating it in the model could be novel. What concerns me is whatever level of novelty there is here it is undermined by lack of detail on the method at which the forecast is included in the ODEs for the SIR model, as I state in my line by line comments it is not clear if you have now an S, alpha, I, R system of ODEs or actually switched to partial differential equations monitoring changes in time and alpha? In addition, the mathematical derivations for these new equations is required before publishing this, at least as a lemma in an appendix.

Major comments:
Line 76: reword making to clarify that it is used to simulate and device control and prevention interventions such as..
Line 90-91: Equation d_alpha/dt, how did you derive this or provide a reference for it; also is this now one of the SIR system of equations, or are you modeling the rate alpha outside the ord. diff. equations - ODE system? And if so, how is this incorporated within the SIR model as you describe in Fig 1C, or did you not intend that? (Also use small capital t as is customary in ODEs)
Fig 2 and 3 depict very simple alteration in the parameters, but it is still not clear if they are outputs of changing these parameters outside the ODE and simply feeding them into the system of ODEs (include the rate of change of alpha) or are you using partial differential equations, or what is going on here?
Line 151: Please provide a reference for the impact of this report, not a peer-reviewed publication, or is this the authors’ opinion?
Line 180: Describe this whole class of models
Line 183: Who is “we”?

Minor comments:
Line 79: spell themselves
Line 155, capitalize Representative Concentration Pathways
Line 162: what do you mean by Murray and others, is this a reference? Shouldn’t this be et al.?
Line 179: The examples shown here represent.. (delete “are”)

Reviewer 1 ·

Basic reporting

No comment.

Experimental design

No comment.

Validity of the findings

No comment.

Additional comments

REVIEW REPORT on
A note on the effects of epidemic forecasts on epidemic dynamics (#48416)

The paper considers an important anecdote on the epidemic dynamics modelling. It examines in what way the epidemic forecasts effect epidemic dynamics via some adaptation on the traditional SIR model. In general, I found the paper appropriate and useful for scholars studying in this research area. Also it fits well to the scope of PeerJ journal. I have only some small points to be corrected before publication. They are as follows:
• In line number 70, in the first equation of the traditional SIR model, “dT" should be “dt” as in other two equations.
• Regarding the Question (2), authors use an infection peak equation. Where this equation come from?
• Authors should briefly provide how they perform transformations from classical SIR to case 1 and case 2. It is beneficial to add what tools they benefit and in what platforms they solve the mathematical problems.

·

Basic reporting

No Comment.

Experimental design

No Comment.

Validity of the findings

No Comment.

Additional comments

In this work, the authors examine how a coupled human system works using a model of viral infection – the susceptible-infected-removed (SIR) model – when the model is used in a forecasting context. Human behavior is modified by making the contact rate responsive to other dynamics, including forecasts, of the SIR system, so that there is two-way feedback between the forecast and the infection dynamics. Results show that a faster rate of response by a population to system dynamics or forecasts leads to a significant decline in peak infections. This is an interesting contribution to the existing literature, but the paper suffers from several shortcomings listed in the following comments.
- The paper should be checked by a native.
- A Conclusion section should be added.
- The abstract’s text should be revised and improved.
- The introduction should be updated by recent researches.
- The novelty and contribution should be clearly bolded.
- In Introduction, the authors should consider some works about different forecasting methods containing regression models and stationary and non-stationary time series models that can be applied to forecast and model different epidemic datasets. For example, they should cite “A novel method to detect almost cyclostationary structure”, “On the detection and estimation of the simple harmonizable processes”, “Periodically correlated modeling by means of the periodograms asymptotic distributions”, “Goodness of fit test for almost cyclostationary processes”, “A new method to compare the spectral densities of two independent periodically correlated time series”, “Testing the difference between spectral densities of two independent periodically correlated (cyclostationary) time series models”, “On the asymptotic distribution for the periodograms of almost periodically correlated (cyclostationary) processes”, “Testing the difference between two independent regression models”, “Testing the equality of two independent regression models”, “On comparing two dependent linear and nonlinear regression models”, “On comparing and classifying several independent linear and non-linear regression models with symmetric errors”, “On comparing, classifying and clustering several dependent regression models”.
- Colure figures are suggested.
- It’s better to suggest some subjects for future works.
Best regards,

---

## Round 0.2 · Minor Revisions

Thanks for addressing the reviewer and my comments, please address the following two comments:

1) Line 86: alpha_star =1 is undefined at this point and reader isn't told the difference from alpha_not = 0.2.
The definitions are listed many lines later (lines 118-9: where you define alpha star as unhindered and alpha not as the reduced contact rate), instead they should be listed soon after the notation.

2) Lines 120-1: This sentence is missing something ("The total pop. size is N=1 in the normalized case"), did you mean frequency dependent transmission with percents rather than population numbers? If so, may be add that for clarity.

Reviewer 1 ·

Basic reporting

No Comment

Experimental design

No Comment

Validity of the findings

No Comment

Additional comments

Authors made a careful revision against my comments.

Therfore, I recommend acceptance of this useful note for PeerJ with its revised version.

Best regards.

·

Basic reporting

no comment

Experimental design

no comment

Validity of the findings

no comment

Additional comments

The title of your article is "A note on the effects of epidemic forecasts on epidemic dynamics".
The readers should know which methods can apply to forecast the epidemic datasets.
These methods are divided in two parts: regression methods and time series methods.
You should say about these techniques.

---

## Round 0.3 · accepted · Accept

Thanks for responding to the reviewers' comments.